# Solvatochromic Sensitivity of BODIPY Probes: A New Tool for Selecting Fluorophores and Polarity Mapping

**DOI:** 10.3390/ijms24021217

**Published:** 2023-01-07

**Authors:** Felix Y. Telegin, Viktoria S. Karpova, Anna O. Makshanova, Roman G. Astrakhantsev, Yuriy S. Marfin

**Affiliations:** 1G.A. Krestov Institute of Solution Chemistry of the RAS, 153045 Ivanovo, Russia; 2Department of Inorganic Chemistry, Ivanovo State University of Chemistry and Technology, 153000 Ivanovo, Russia; 3Department of Natural Sciences, Mendeleev University of Chemical Technology of Russia, 125047 Moscow, Russia; 4HSE Tikhonov Moscow Institute of Electronics and Mathematics, HSE University, 101000 Moscow, Russia

**Keywords:** BODIPY, fluorescent probe, absorption, emission, solvatochromic sensitivity, biosensing

## Abstract

This research work is devoted to collecting a high-quality dataset of BODIPYs in a series of 10–30 solvents. In total, 115 individual compounds in 71 solvents are represented by 1698 arrays of the spectral and photophysical properties of the fluorophore. Each dye for a series of solvents is characterized by a calculated value of solvatochromic sensitivity according to a semiempirical approach applied to a series of solvents. The whole dataset is classified into 6 and 24 clusters of solvatochromic sensitivity, from high negative to high positive solvatochromism. The results of the analysis are visualized by the polarity mapping plots depicting, in terms of wavenumbers, the absorption versus emission, stokes shift versus − (absorption maxima + emission maxima), and quantum yield versus stokes shift. An analysis of the clusters combining several dyes in an individual series of solvents shows that dyes of a high solvatochromic sensitivity demonstrate regular behaviour of the corresponding plots suitable for polarity and viscosity mapping. The fluorophores collected in this study represent a high quality dataset of pattern dyes for analytical and bioanalytical applications. The developed tools could be applied for the analysis of the applicability domain of the fluorescent sensors.

## 1. Introduction

Currently, the solvatochromic properties of BODIPY sensors and other efficient probes are being extensively explored in analytical [1,2], bio, and medicinal chemistry [3,4,5,6,7] due to their high sensitivity and their selectivity to environmental macro- and micro-polarity. A wide variety of BODIPYs and other fluorophores are reported in scientific papers and a design strategy for future applications is developed [8,9,10,11,12]. Numerous examples have recently been reviewed for polarity sensitive probes of various subcellular structures [13,14,15,16], reactive oxygen and nitrogen species [17,18], viscosity sensitive probes based on the rotor effect [19,20,21] and AIE-effect [22,23,24], polarity-controlled triplet photosensitizers for photodynamic therapy [25,26,27], photoswitches, and photocages for theranostic applications [28,29,30,31,32], among others. In some cases, the insensitivity of BODIPY sensors to solvent polarity makes them ideal for bioimaging [15,33,34].

Routine information regarding the spectral and photophysical properties of dyes important for the further development of BODIPY fluorophores and discussions regarding their structure–property relationships are dispersed in numerous papers and reviews that have been published within the last two decades. In the original research, several libraries of BODIPY for intracellular cell imaging have been reported for individual solvents, namely: dimethylsulfoxide [35,36,37,38,39,40], methanol [41,42], and HEPES buffer solution [43]. Much of the information summarizing and arranging the properties of the fluorophores of diverse structures, including BODIPY dyes, in various solvents is provided in the extensive experimental databases, regarding the optical and photophysical properties needed for analyses [44,45,46,47,48].

One of the key strategies for the design and optimization of fluorescent sensors is focused on the polarity mapping of biological tissue. Several structures based on BODIPY scaffolds exemplify selected polarity probes developed in numerous research works [8,49,50,51,52,53,54]. Other prevalent scaffolds such as coumarin, xanthene, cyanine, pyrene, naphthalene, carbazole, porphyrin, phthalocyanine, and some others are also intensively discussed [15,55].

The above libraries present useful data for analysis and for seeking target molecules satisfying the requirements of the spectral and photophysical properties, sensitivity to surrounding media, selectivity for adsorption on biological tissue, etc. An examination of the above libraries shows a lack of data for selecting BODIPY fluorophores, which could serve as analytical templates for polarity.

Recent progress in the quantification of the solvatochromism of BODIPY probes [56,57] is promising for the further development of polarity-based approaches for tailoring the structure of fluorophores. Data of such kind have not been systematically analyzed in the literature. Therefore, the current research aims at collecting high quality, low scale datasets for the analysis of the spectral absorption and emissions of BODIPY fluorophores, as well as the quantum yield in a long series of solvents (above 10) within a wide range of polarities, and a corresponding analysis of the effects in terms of the relationship of the spectral properties and the solvatochromic sensitivity of the compounds.

## 2. Results

### 2.1. Overview of the Properties of Solvatochromic BODIPYs and Solvents Explored in the Dataset

The list of fluorescent dyes selected for the research is shown in Table 1. 

A general overview of the spectral properties, Stokes shift, and quantum yield for the whole dataset is demonstrated by the frequency diagrams in Figure 1. The properties correspond to the whole range of 71 solvents depicted in Appendix A.

### 2.2. Cluster Analysis of the Solvatochromic Sensitivity

#### 2.2.1. Six-Cluster Dataset of High Solvatochromic Sensitivity 

The total dataset of 115 BODIPY dyes and 1698 combinations of dye and a solvent was divided into six clusters according to the solvatochromic sensitivity parameter A. The solvatochromic sensitivity (or polarity) of each compound in the whole dataset of BODIPY sensors varied in the range of −1.3 to +1.2. Each cluster combined several series of dyes, and a series of solvents for each dye appeared only once, according to the arrangements of the data. 

Parameter A, representing solvatochromic sensitivity, was defined according to a method described in Section 3.3, through the use of the absorption and emission maxima of the fluorophore in a series of solvents.

The chemical structures of the BODIPY sensors classified according to the above procedure of clustering are provided in the six chats below (Figure 1, Figure 2, Figure 3, Figure 4, Figure 5 and Figure 6).

The spectral and photophysical properties of the dyes in the clusters are presented below in several diagrams. Figure 2 shows the relationship of the absorption maxima versus the emission maxima according to Equation (4) in terms of nm. Figure 3 demonstrates the relationship of the Stokes shift versus − (Absorption maxima + Emission maxima), which corresponds to the basic Equation (1) and is expressed in 1000/cm. 

The clustering of the dyes according to solvatochromic sensitivity parameter A helps understand the general concept of the solvatochromic classification of the dyes. For instance, the plot for absorption maxima versus emission maxima for cluster 1 in Figure 2 and Figure 3 exhibiting high positive solvatochromism depicts several vertically oriented asterisms with an approximately constant level of absorption in the solvents of a different nature, while the emission properties are varied in a wide range, indicating the high polarity of the molecule in the excited state. On the contrary, the absorption versus emission plot for cluster 6 demonstrates a wide variation in the absorption maxima and a low variation of emission characterizing the low polarity of the molecules in the excited state. Such behavior for the fluorophore is not typical. The increased values of the Stokes shift in cluster 1 in Figure 4 and Figure 5 are shifted to the positive direction of the x-axis (less negative values), while those in cluster 6 to the opposite side follow the values of parameter A. If we take into consideration the other two pairs of symmetric clusters 2, 5, and 3, 4, shown in Figure 4 and Figure 5, respectively, the characteristic behavior of the less polar dyes is similar to clusters 1 and 6, but not much more pronounced.

The frequency diagram shown in Figure 6 demonstrates a high quantum yield for a large number of dyes bearing near-zero solvatochromism (clusters 4 and 5), indicating the role of the dye polarity in the quantum yield. The plot of toe quantum yield vs Stokes shift, shown in Figure 7, is convenient for practical use. 

#### 2.2.2. 24-Cluster Dataset of High Solvatochromic Sensitivity

More detailed clustering of the dataset up to count 24 helps to identify the more regular behavior of the dyes in several clusters. The first one, the absorption maxima versus the emission maxima, is shown in Figure 8, for an approximate visualization of the absorption and the emission spectral range of the BODIPY fluorophores. Figure 8 illustrates the general semiempirical rule for the relationships between the absorption and emission wavenumbers approximated by Equation (4). Figure 9 clarifies the role of the relationship between the Stokes shift and − (Absorption maxima + Emission maxima) for polar sensors according to Equation (1). The plot of the quantum yield versus Stokes shift in Figure 10 supports understanding a common statement for polarity sensitive fluorophores—the decrease in quantum yield is accompanied by an increase in the Stokes shift.

The diagrams in Figure 9 and Figure 10 (see in particular clusters 1–2 and 23–24) show regular behavior for dyes bearing a high positive and high negative solvatochromic sensitivity, respectively. Those fluorophores exhibiting a charge transfer in polar solvents are suitable for polarity mapping due to the high deviation in the absorption and emission wavelength, as well as the quantum yield and Stokes shift. The chemical structures of the corresponding dyes are shown below in Figure 7 and Figure 8. Several compounds in this multitude coincide with those estimated in chemoinformatics research based on the submolecular fragments approach [57].

It is noteworthy that the diagram for quantum yield versus Stokes shift in Figure 10 for several dyes in clusters 1, 2, and 23, 24 could be approximated by one curve, respectively, while the plot for the Stokes shift versus − (absorption maxima + emission maxima) for the correspondent clusters in Figure 9 could be approximated by two lines. As an exception, a slight increase in the quantum yield versus Stokes shift is exhibited for parent aza-BODIPY, 2011 Banuelos(3437)-BTAA [69], which is provided in the dataset for comparison with the dipyrrin dyes. A similar behavior is demonstrated by 2018 Leen-BODIPY [93] in cluster 22/24, which could be attributed to the keto-enol tautomerism of the diketone fragment of the molecule.

#### 2.2.3. BODIPYs Bearing Near-Zero Solvatochromism

The diagrams in Figure 10 reveal several groups of dyes exhibiting near-zero solvatochromic sensitivity. Several dyes in cluster 18 bearing near-zero solvatochromism form a single group approximating the relationship of quantum yield versus Stokes shift in a wide range of Stokes shift for fluorescence emission within 530–630 nm. The dyes are shown in Figure 9.

The individual dyes covering a wide range of quantum yields are exemplified by the following compounds shown in Figure 10.

The above-mentioned members of clusters 7, 8, 9, 10, and 19 formed groups of dyes with quantum yields within 0.02–1 and Stokes shifts within 170–860 cm^−1^, 1270–1400 cm^−1^, and 2190–2800 cm^−1^. 

Several dyes mentioned above in this section exhibited molecular rotor effects, for example, those in clusters 7a, 9a, 9b, and 19. The dye in cluster 7 is known to exhibit a high quantum yield in solvents with a high polarity. This observation demonstrates the applicability of the developed tool for selecting efficient fluorophores demonstrating a low spectral sensitivity and high sensitivity of quantum yield to the polar solvents.

### 2.3. Analysis of Duplicates and Evaluation of Deviations of Spectral Data

A new tool for selecting the fluorescent probes proposed above is convenient for the analysis of deviation of experimental data for identical compounds. In essence, the problem of duplicates in chemoinformatics is solved by special algorithms based on, for example, SMILES or IUPAC InChI formats [107]. Normally, duplicate chemical structures are deleted from the dataset. However, spectral data for the essayed repeated compounds, as well as their solvatochromic sensitivity, could be used for evaluating the deviation of the correspondent properties in various experiments.

The chemical structures of the duplicates of the dataset collected in the current research are represented in Figure 11. The deviation of the solvatochromic sensitivity parameter A for each duplicate is shown in Figure 11 and Figure 12. The solvent count is shown above each column. All of the cases with a solvent count equal to or above 10 correspond to the dataset in Table 1, while a solvent count below 10 corresponds to the extra dataset of duplicates shown in Table 2, which were collected from the literature for a comparison of the spectral data.

The analysis of the solvatochromic sensitivity parameter A for all duplicates is a compact form of comparison of the data for solvatochromic behaviour compared with the whole spectral dataset. The solvatochromic sensitivity of the compounds of the groups G, M, N, O, P, Q, and R lies within the limits of the experimental error shown in the dataset represented in files of SI.csv and SI.sdf. However, several other compounds depict a high deviation, for example, a parent compound 2009 Arroyo-1 in group A, 2015 Waddell-1, 2015 Waddell-2, and the compounds in groups I and J. 

It is noteworthy that several duplicate compounds have been studied in a number of experiments for quite different solvents. For example, the dyes in group F demonstrate a similar solvatochromic sensitivity with a positive solvatochromism (A > 0) for a series of 16, 22, and 6 solvents. 

According to the data in the diagram shown in Figure 1, the mean standard deviation SD and standard deviation of the mean SD/√n for the solvatochromic sensitivity are estimated as follows, respectively: group A: (with an excluded extra value for 2009 Aroyo-1) 0.077, 0.037; group B: 0.091, 0.037; group C: 0.076, 0.031.

The above results could be clarified by a comparison of the spectral and photophysical data reported by different authors for each solvent. A few examples are shown for groups A, B, C, and D, respectively, in the figures, namely: Appendix A. The tooltip text for each mark in the figure demonstrates the compound ID, solvent name, solvatochromic sensitivity, relation E/C, absorption maxima, emission maxima, Stokes shift, and quantum yield.

## 3. Materials and Methods

### 3.1. Dataset of BODIPYs

The list of dyes included a total dataset of 115 BODIPYs and the corresponding references are shown above in Table 1 (Section 2.1). The dataset of BODIPY fluorophores collected in this study are listed in SI.csv and SI.sdf files with a correspondent count for solvents; references and IDs generated from the year of publication, first author, and title of the compound in the paper. The chemical structure in SMILES format and the direct code of JChem [107] was used for arranging the data. Each series of spectral and photophysical data are represented by 10–30 solvents for the analysis of the solvatochromic effect.

The chemical structures of the fluorophores are depicted in Figure 1, Figure 2, Figure 3, Figure 4, Figure 5 and Figure 6 in Section 2.2.1.

### 3.2. Evaluation of Statistical Deviations in Spectral and Photophysical Data

The dispersion of spectral data was evaluated through the use of duplicates in the dataset. For this purpose, additional data for spectral behaviour of BODIPYs covering six to nine solvents in each series, shown in Table 2 (Section 2.3) (see also Appendix A), were used along with software for the analysis and modelling of the molecular properties, such as DataWarrior [135] and ChemMine Tools [136].

### 3.3. Calculation of Solvatochromic Sensitivity of BODIPY Fluorophores

In previous research, a novel methodology was reported for studying the solvatochromism of BODIPYs [56,57]. This approach explores the empirical generalization of solvents’ functions in Liptay’s theory [137] of solvatochromism yielding an easy approach for treating a wide variety of spectral data published in the literature. The corresponding semiempirical equations are expressed as follows:(1)$$\nu \left(Abs\right)-\nu \left(Em\right)=-A\left[\nu \left(Abs\right)+\nu \left(Em\right)\right]+B$$
(2)$$\nu \left(Abs\right)=\frac{1-A}{2}\left[\nu \left(Abs\right)+\nu \left(Em\right)\right]+\frac{B}{2}$$
(2a)$$\nu \left(Abs\right)=-C\left[\nu \left(Abs\right)+\nu \left(Em\right)\right]+D$$
(3)$$\nu \left(Em\right)=\frac{1+A}{2}\left[\nu \left(Abs\right)+\nu \left(Em\right)\right]-\frac{B}{2}$$
(3a)$$\nu \left(Em\right)=-E\left[\nu \left(Abs\right)+\nu \left(Em\right)\right]+F$$
(4)$$\nu \left(Em\right)=\frac{1+A}{1-A}\nu \left(Abs\right)-\frac{B}{1-A}$$
(5)$$A=\frac{{\left(\mu_{e}-\mu_{g}\right)}^{2}}{\mu_{e}^{2}-\mu_{g}^{2}}$$
where $\nu \left(Abs\right)$, $\nu \left(Em\right)$ is the wavenumbers that correspond to the absorption and emission maxima; *A, B* are the solvatochromic sensitivity coefficients, evaluated as regression coefficients of the linear approximation $\nu \left(Abs\right)-\nu \left(Em\right)$ vs $-\left[\nu \left(Abs\right)+\nu \left(Em\right)\right]$; and *µ*(*g*) and *µ*(*e*) are the vectors of the dipole moments of the fluorophore in the ground and excited states. For simplicity, the vector symbol is omitted.

Parameter *A,* described by Equation (5), is written in a more general form than in previous research [56,57] (see Appendix A). It helps to use dipole moments in vector form with no additional assumptions because the square of a vector function has a scalar value.

As far as parameter *A* is a function of dipole moments controlling the solvation of the fluorophore and dye spectral behavior in the ground and excited states, it is reasonable to use it as a quantitative measure of the solvatochromic sensitivity.

Coefficient *A* can be experimentally evaluated from the regression coefficients of Equations (1), (2a), and (3a).


The form of Equations (1), (2a), and (3a) is convenient for the simultaneous analysis of the behaviour of the three spectral functions, $\nu \left(Abs\right)-\nu \left(Em\right)$, $\nu \left(Abs\right)$, and $\nu \left(Em\right)$, shown in one plot (see Figure 13, Figure 14 and Figure 15 below).

A pairwise comparison of Equations (2) and (2a), and (3) and (3a) yields slopes *C* and *E*, respectively:(6)$$C=-\frac{1-A}{2};\;E=-\frac{1+A}{2}$$

Solvatochromic parameter *A* is estimated directly as the regression coefficient of Equation (1) for the Stokes shift or as the difference between *C* and *E*:(7)A=C−E

On the other hand, parameters *C* and *E*, defined by Equation (6), are suitable for estimating the relationship $\nu \left(Abs\right)$ vs $\nu \left(Em\right)$:(8)$$\frac{1+A}{1-A}=\frac{E}{C}$$
in order to apply Equation (4).

Expression (7) is useful for the measurement of the solvatochromic sensitivity of the relatively high positive and negative values, *A *> 0, *A* < 0, while the relation *E*/*C*, defined by Equation (8), is useful for characterising the near-zero solvatochromic sensitivity, *A* ≈ 0, *E*/*C* ≈ 1.

In the case of coplanarity of the vectors of the dipole moments *µ*(*g*) and *µ*(*e*), they can be used as scalar parameters, thus Expressions (5) and (8) are simplified for a clear understanding of the physicochemical meaning of parameter *A* defining the solvatochromic sensitivity:(9)$$A=\frac{\mu \left(e\right)-\mu \left(g\right)}{\mu \left(e\right)+\mu \left(g\right)},\;\;\frac{E}{C}=\frac{1+A}{1-A}=\frac{\mu \left(e\right)}{\mu \left(g\right)}$$

An illustration of the calculation method is provided in the papers [56,57]. Some additional examples are given below in Figure 13, Figure 14 and Figure 15 for the fluorophores discussed above. In the case of high positive and negative solvatochromism (Figure 13 and Figure 14), the Stokes shift exhibits a low statistical deviation from the linear correlation suitable for the estimation of parameter A; in the case of near-zero solvatochromism (Figure 15), the absorption or emission wavenumbers are more suitable for this purpose. 

The experimentally evaluated parameters of Equations (1), (2a), (3a), (7), and (8), as well as their statistical characteristics for a dataset of 115 BODIPYs, are provided in the Appendix A.

### 3.4. Clustering the Properties of the Fluorophores and the Graphical Analysis

The clustering of the experimental data for a whole family of BODIPYs was performed by the parameters of solvatochromic sensitivity parameter A. This method of presentation helps to highlight each dye in a series of solvents, as well as to compare different series with a similar solvatochromic sensitivity. A graphical analysis of the spectral and photophysical properties of the collected BODIPYs was performed with the software Scimago Graphica [138].

## 4. Conclusions

Based on published results, a dataset of 115 BODIPY sensors in a series of 10–30 solvents with correspondent spectral and photophysical characteristics was collected. The solvatochromic sensitivity of each dye was evaluated by a recently developed semiempirical approach for the analysis of the solvatochromism. The polarity of the dyes was divided into 6 and 24 clusters from high negative to high positive levels, and was applied for the classification of fluorophores and an analysis of their properties.

The analysis of clusters combining several dyes in an individual series of solvents shows that dyes with a high polarity, with either a positive or negative solvatochromism, demonstrated regular changes in the absorption maxima versus emission maxima, Stokes shift versus − (absorption maxima + emission maxima), and quantum yield versus Stokes shift in a wide range of properties. Non-polar probes were arranged into characteristic clusters with quantum yields of 0.02–1 and Stokes shifts of 170–860 cm^−1^, 1270–1400 cm^−1^, and 2190–2800 cm^−1^.

The developed tool based on the classification of solvatochromic sensitivity is suitable for the efficient selection of fluorescent probes, for both the polarity and viscosity sensing of biological tissue through the use of individual dyes and a set of dyes. The selected BODIPY probes could serve as pattern samples for the testing and design of the fluorophores.

## Data Availability

https://doi.org/10.6084/m9.figshare.21805410.v1.

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
