# Peer review of "Solvatochromic Sensitivity of BODIPY Probes: A New Tool for Selecting Fluorophores and Polarity Mapping"

_ijms, 2023, doi:10.3390/ijms24021217_

Round 1

Reviewer 1 Report

The F.Y. Telegin et al.’s paper concerns with a collection of high quality dataset of spectral and photophysical properties of BODIPYs treated by semiempirical approaches to highligh the solvatochromic sensitivity of the compounds.

The study is interesting and the obtained results could be newsworthy for applications in different fields such as fluorescent sensors and bioanalyses.

However, the manuscript is difficult to read and understand, the results are unclearly presented, key references are missing, and the meaning of some parameters is implied and not clearly defined. For these reasons the paper cannot be published in the present form but needs a thorough revision.  

1)      The main point is the meaning of the transion energy E00. The latter is the energy difference between the 0 point (vibrational level =0) of the excited state and the 0 point (vibrational level =0) of the ground state, in agreement with the calculations proposed in ref. 32 and cited on p.20 line 4 from the bottom. E00 can be estimated through the intersection (not the intercept) of the normalized absorption and emission spectra, as reported on p. 20 line 18.  I wonder why the same symbol is also used to define a different quantity (the negative sum of absorption and emission maxima), as stated in the abstract (line 8).

2)      The theory applied in the present paper and the derived parameters have to be better clarified. In fact, when citing the Liptay’s theory of solvatochromism (p. 20 line 15) no references have been mentioned. It seems to this reviewer that the values calculated for A and B parameters in Equation 1 are not shown in the text and poorly discussed.

3)      Table S1 , showing the BODIPYs structures, has to be introduced at the biginning of the Results section. Moreover, replace Table S5 on p. 2 line 18 by the correct number (Table S3) .

4)      The acronym (AE-Plot), first introduced at the last line of p. 2, is not explained at all, nor is it referred to another section of the manuscript in which it is explained.

5)      On p. 3 line 2, the sign - is inside the brackets while in equation 1 it is outside, which one is correct?

6)      The parameter A for the solvatochromic sensitivity is introduced for the first time on p. 3 line 7 and it is neither explained nor referenced to equations in the text.

7)      the ordinate and abscissa together with the units of measurement must always be shown in the figures and not only mentioned in the captions of Figures. Moreover, in the caption of Figure 7 replace cm by cm-1.

8)      First line of p. 14: probably the authors have to add “see in particular” before  clusters 1-2 and 23-24 in the parentheses.

9)      Figure 9, introduced at the beginning of section 2.3, has to be better explained .

10)   At the end of Scheme 10 are shown 6 BODIPYs structures neither numbered nor mentioned in the caption.

11)   P. 17, line 3: the Stokes shift values are usually given as cm-1 and not nm.

12)   The relation A(Em)/A(abs), first mentioned on p. 17 line 5 from the bottom, has to be better introduced and defined.

Reviewer 2 Report

F.Y. Telegin et al reported an article on the solvatochromic sensitivity of several series of BODIPY probes. BODIPY probes were categorized into different series based on their polarity and photophysical properties were differentiated apparently. This is an important work and provides instruction on designing BODIPY probes in far-red/near IR range. Acceptance after minor revision is recommended. Several recent references are suggested to be cited below.

1. Photo-uncaging of BODIPY oxime ester for histone deacetylases induced apoptosis in tumor cells. Chem. Commun., 2019,55, 14162-14165

2. Efficiency of Functional Group Caging with Second-Generation Green- and Red-Light-Labile BODIPY Photoremovable Protecting GroupsJ. Org. Chem. 2022, 87, 21, 14334–14341

3. BODIPY-Based Photoacid Generators for Light-Induced Cationic Polymerization.Org. Lett. 2020, 22, 3, 1208–1212

4. Controlling Antimicrobial Activity of Quinolones Using Visible/NIR Light-Activated BODIPY Photocages. Pharmaceutics 2022, 14(5), 1070

5. “Turn on” orange fluorescent probe based on styryl-BODIPY for detection of hypochlorite and its application in live cell imaging. Dyes and Pigments, 2019, 162, 189-195

6. Synthesis and bioimaging of a BODIPY-based fluorescence quenching probe for Fe3+. RSC Adv., 2022,12, 21332-21339

7. A BODIPY-Based Far-Red-Absorbing Fluorescent Probe for Hypochlorous Acid Imaging. ChemPhotoChem 2022, 6 (4), e202100250

Round 2

Reviewer 1 Report

“…the manuscript is difficult to read and understand, the results are unclearly presented, key references are missing, and the meaning of some parameters is implied and not clearly defined. For these reasons the paper cannot be published in the present form but needs a thorough revision.”

My main comment to the paper unfortunately remains unchanged. The authors, in an attempt to correct and improve the paper, inserted typing errors showing that they had not carefully checked the manuscript before resubmitting it. This makes it even more difficult to read and understand the paper.

As examples:

1.       Line 46: replace Figure 2 by Figures 2 and 3

2.       Line 47: replace Figure 3 by Figures 4 and 5

3.       Line 73: delete the old caption of Figure 3

4.       Schemes 7, 8, 9 and 10 are poorly laid out and illegible. Moreover, the captions to the same Schemes are inserted several times. Please correct.

5.       Line 167: replace Figure 8 by Figure 9

6.       Line 169: replace Figure 9 by Figure 10

7.       Line 286: replace Figure 8 by Figure 10

8.       Line 289: replace Figure 8 by Figure 9

9.       In lines 303-306 is reported the same sentence in lines 299-302, please delete it.

10.   Line 396: I assume that there are three (and not two) groups identified since line 398 shows three Stokes shift ranges.

11.   Caption to Figure 11: The old notation A(Em)/A(Abs) has been used, please correct.

Major concerns:

1.       “The first point is the meaning of the transion energy E00. The latter is the energy difference between the 0 point (vibrational level =0) of the excited state and the 0 point (vibrational level =0) of the ground state, in agreement with the calculations proposed in ref. 32 and cited on p.20 line 4 from the bottom.”

I agree with your definition on line 539 ?00 = ½ [?(???) + ?(??)]. The same quantity can be calculated as the intersection of the normalized absorption and emission spectra.

-          Please, clarified the sentence on lines 536.538 “Half of this 536 quantity is evaluated from the interception of absorption and emission plots which is char- 537 acterising the sum of vertical transition energies of absorption and emission”  and  that on lines 502-503 “and intercept of normalized absorption and emission (E00 energy)”. Moreover, delete “(transition energy E00)” in the abstract.

2.       “The theory applied in the present paper and the derived parameters have to be better clarified.”

It is not clear as some equations are derived, please insert suitable references just for equations 5 and 10.

Moreover, it is not clear how the reported parameters E/C have been calculated, in fact the figures in the text report the Stokes shift as a function of –(Absorption maxima + Emission maxima). The slope of the linear regression of  ?(???) − ?(??) vs –[?(???) + ?(??)] should directly give the A parameter, therefore I do not understand why the parameters E and C have been introduced. Please clarify as the parameters E/C reported on the Figures 11 and 12 have been calculated, also showing one linear regression with the correlation coefficient and the derived parameters.

-           

Author Response

Response to Reviewer 1

I am grateful for the extremely useful comments of Reviewer 1 regarding the scientific value of the manuscript and editorial details.

We have received kind permission from the editor to accept the previous changes to make further revisions to the manuscript and to avoid the difficulties of understanding numerous changes.

Please find below our comments on the changes to the revised manuscript.

Reviewer:

Comments and Suggestions for Authors

“…the manuscript is difficult to read and understand, the results are unclearly presented, key references are missing, and the meaning of some parameters is implied and not clearly defined. For these reasons the paper cannot be published in the present form but needs a thorough revision.”

My main comment to the paper unfortunately remains unchanged. The authors, in an attempt to correct and improve the paper, inserted typing errors showing that they had not carefully checked the manuscript before resubmitting it. This makes it even more difficult to read and understand the paper.

As examples:

  1. Line 46: replace Figure 2 by Figures 2 and 3
  2. Line 47: replace Figure 3 by Figures 4 and 5
  3. Line 73: delete the old caption of Figure 3
  4. Schemes 7, 8, 9 and 10 are poorly laid out and illegible. Moreover, the captions to the same Schemes are inserted several times. Please correct.
  5. Line 167: replace Figure 8 by Figure 9
  6. Line 169: replace Figure 9 by Figure 10
  7. Line 286: replace Figure 8 by Figure 10
  8. Line 289: replace Figure 8 by Figure 9
  9. In lines 303-306 is reported the same sentence in lines 299-302, please delete it.

Response:

We have carefully checked all the references for figures, schemes, and equations in the text. All the duplicate sentences are deleted.

An additional table (Table 2) is provided as a reference for the duplicate compounds (line 161).

A comment regarding the standard deviation of the duplicates is provided in the text (lines 174-177).

Editing of the text is performed.

Reviewer:

  1. Line 396: I assume that there are three (and not two) groups identified since line 398 shows three Stokes shift ranges.

Response:

Corrected in the text of the manuscript, the Abstract and Conclusions.

Reviewer:

  1. Caption to Figure 11: The old notation A(Em)/A(Abs) has been used, please correct.

Response:

Figures 11-12 are changed from the relation E/C to solvatochromic sensitivity A. This helps to make a discussion according to one concept.

Reviewer:

Major concerns:

  1. “The first point is the meaning of the transion energy E00. The latter is the energy difference between the 0 point (vibrational level =0) of the excited state and the 0 point (vibrational level =0) of the ground state, in agreement with the calculations proposed in ref. 32 and cited on p.20 line 4 from the bottom.”

I agree with your definition on line 539 ?00 = ½ [?(???) + ?(??)]. The same quantity can be calculated as the intersection of the normalized absorption and emission spectra.

-          Please, clarified the sentence on lines 536.538 “Half of this 536 quantity is evaluated from the interception of absorption and emission plots which is char- 537 acterising the sum of vertical transition energies of absorption and emission”  and  that on lines 502-503 “and intercept of normalized absorption and emission (E00 energy)”. Moreover, delete “(transition energy E00)” in the abstract.

Response:

The text regarding transition energy is deleted as far as we do not provide new knowledge in this area. In the revised state the manuscript is focusing directly on (Absorption maxiam+Emission maxima) as a function for characterising the solvatochromic sensitivity based on the developed semiempirical approach. The Reviewer’s notes will be carefully studied for more clear explanation of our ideas in future research.

Noteworthy, part of the deleted text contains several references, therefore we did not use tracking of the changes. A copy of the deleted text is presented below;

  1. “The theory applied in the present paper and the derived parameters have to be better clarified.”

It is not clear as some equations are derived, please insert suitable references just for equations 5 and 10.

Moreover, it is not clear how the reported parameters E/C have been calculated, in fact the figures in the text report the Stokes shift as a function of –(Absorption maxima + Emission maxima). The slope of the linear regression of  ?(???) − ?(??) vs –[?(???) + ?(??)] should directly give the A parameter, therefore I do not understand why the parameters E and C have been introduced. Please clarify as the parameters E/C reported on Figures 11 and 12 have been calculated, also showing one linear regression with the correlation coefficient and the derived parameters.

-           

Response:

(Lines 225-231, 241-261): Section 3.3 is rearranged, which makes it possible to describe in more detail the experimental estimation of the solvatochromic sensitivity parameter and to make a graphical presentation of solvatochromism in 3 additional Figures 13-15.

Besides this, some additional notes regarding the semiempirical approach for quantifying the solvatochromism are provided in the supplementary information SI.docx.

The references (56, 57) for our previous research are provided in the text of the manuscript.

Two accidental misprints in equations (6) and 7) are removed.

Round 3

Reviewer 1 Report

The authors corrected the manuscript along the lines of my comments and suggestions.

One last minor change remains before the paper can be published in IJMS. I believe that in equations 6 and 7 the ”–“ sign is wrong and “+C” and “+E” must be inserted, respectively, in accordance with the equations 2 and 3 and the values of the slopes of the linear correlations shown in figures 13 and 14.

Author Response

Response to Reviewer 1

We are grateful for the numerous details suggested by Reviewer 1 during the whole reviewing process. All of them have principal value for the quality of the manuscript and demonstrate a deep understanding of the materials of our research.

Reviewer:

One last minor change remains before the paper can be published in IJMS. I believe that in equations 6 and 7 the ”–“ sign is wrong and “+C” and “+E” must be inserted, respectively, in accordance with the equations 2 and 3 and the values of the slopes of the linear correlations shown in figures 13 and 14.

Response:

A respected Reviewer is absolutely right! This is our mistake made a moment before sending the response as a result of a strong desire to clarify the question.

It goes without saying that we should use +C and +E in equations 6 and 7, respectively.

On the other hand, if we use those equations in a present form, then we find the difficulty of comparative analysis of the behavior of 3 plots in Figures 13-15 because we need to change the sign of the coefficients and arguments in our mind.

Therefore, we take a chance to make the last changes of Section 3.3 for a clear analysis of the plots in Figures 13-15 and simultaneous calculation of the regression coefficients.

For easy tracking of the changes, we place a corrected text of the previous version with +C and +E below

We propose to change the sign of the coefficients defined by equations (6) below and as a result to change equation (7). Those equations as mentioned by Reviewer 1 satisfy correspondent equations (2) and (3). In this case, the coefficients C and E reflect the real value estimated from the regression shown in Figures 13-15 and Supplementary Information. Values of C and E are added to the captures in Figures 13-15. It is worth reminding here that the form of equation (1) or (1’) corresponds to A>0 for positive solvatochromism when according to equation (5) the dipole moment in the excited state is higher than that in the ground state, and A<0 for the opposite case. The important changes are highlighted in yellow.

Lines 232-261: The revised text of this part is as follows.

Coefficient A can be experimentally evaluated from the regression coefficients of the following equations (1’)-(3’):

                (1’)

                        (2’)

                         (3’)

The form of equations (1’)-(3’) is convenient for simultaneous analysis of the behavior of three spectral functions ,  and  shown in one plot (see Figures 13-15 below).

A pairwise comparison of the equations (2) and (2’), (3) and (3’) yields the slopes C and E :

;                                   (6)

Solvatochromic parameter A is estimated directly as the regression coefficient of equation (1’) for Stokes shift or as a difference between C and E:

                                                     (7)

On the other hand, the parameters C and E defined by Equation (6) are suitable for estimating the relationship  vs :

                                                          (8)

to apply Equation (4).

Expression (7) is useful for the measurement of solvatochromic sensitivity of relatively high positive and negative values, A>0, A<0, while the relation E/C defined by Equation (8) is useful for characterising near-zero solvatochromic sensitivity, A»0, E/C»1.

In the case of coplanarity of the vectors of dipole moments µ(g) and µ(e), they can be used as scalar parameters, therefore expressions (5) and (8) are simplified for a clear understanding physicochemical meaning of the parameter A defining solvatochromic sensitivity:

,                        (9)

An illustration of the calculation method is provided in the papers [56,57]. Some additional examples are given below in Figures 13-15 for the fluorophores discussed above. In the case of high positive and negative solvatochromism (Figures 13 and 14) Stokes shift exhibits low statistical deviation from the linear correlation suitable for estimation of the parameter A, in the case of near-zero solvatochromism (Figure 15) the absorption or emission wavenumbers are more suitable for this purpose.

Experimentally evaluated parameters of Equations (1’)-(3’), (7), and (8) as well as their statistical characteristics for a dataset of 115 BODIPYs are provided in the Supplementary Information.
